# Quasi Pd$_1$Ni single-atom surface alloy catalyst enables hydrogenation of nitriles to secondary amines

Hengwei Wang[1,2,4], Qiquan Luo [1,4], Wei Liu[3], Yue Lin [1], Qiaoqiao Guan[1,2], Xusheng Zheng[3], Haibin Pan[3], Junfa Zhu [3], Zhihu Sun [3], Shiqiang Wei [3], Jinlong Yang [1,2] & Junling Lu [1,2]*

Hydrogenation of nitriles represents as an atom-economic route to synthesize amines, crucial building blocks in fine chemicals. However, high redox potentials of nitriles render this approach to produce a mixture of amines, imines and low-value hydrogenolysis byproducts in general. Here we show that quasi atomic-dispersion of Pd within the outermost layer of Ni nanoparticles to form a Pd$_1$Ni single-atom surface alloy structure maximizes the Pd utilization and breaks the strong metal-selectivity relations in benzonitrile hydrogenation, by prompting the yield of dibenzylamine drastically from ~5 to 97% under mild conditions (80 °C; 0.6 MPa), and boosting an activity to about eight and four times higher than Pd and Pt standard catalysts, respectively. More importantly, the undesired carcinogenic toluene by-product is completely prohibited, rendering its practical applications, especially in pharmaceutical industry. Such strategy can be extended to a broad scope of nitriles with high yields of secondary amines under mild conditions.

[1] Hefei National Laboratory for Physical Sciences at the Microscale, University of Science and Technology of China, Hefei 230026, P. R. China. [2] Department of Chemical Physics, Key Laboratory of Surface and Interface Chemistry and Energy Catalysis of Anhui Higher Education Institutes, iChem, University of Science and Technology of China, Hefei 230026, P. R. China. [3] National Synchrotron Radiation Laboratory, University of Science and Technology of China, Hefei 230029, P. R. China. [4]These authors contributed equally: Hengwei Wang, Qiquan Luo. *email: junling@ustc.edu.cn

Among nitrogen-containing chemicals, amines are important and ubiquitous in various biological active compounds[1,2], and are also valuable building blocks for synthesis of polymers, dyes, pharmaceuticals, agrochemicals, and fine chemicals in industry[3–7]. Compared with other functional compounds, synthesis of amines has received one of the most extensive attentions in organic chemistry[3,8]. Several methods such as amination of aryl and alkyl halides[9–11], reductive amination of aldehydes and ketones[12,13], amination of alcohols[7,14] and hydroaminations of olefins[6,15], have been developed to construct the carbon–nitrogen bonds for amines synthesis. However, these routes require either base additives or high cost, and also produce heavy liquid wastes in general. Alternatively, hydrogenation of readily available nitriles using molecular hydrogen over heterogeneous[16–23] and homogeneous[5,24–26] metal catalysts has been recognized as a more environmentally benign and atom-economic route to synthesis of these value-added amines. While, owing to the high redox potentials of nitriles, this process often shows severe selectivity issues, and generally produces mixtures of primary, secondary, and tertiary amines, imines and low-value hydrogenolysis by-products[4,27], which makes this approach costly in the sequential products separation due to small differences in their boiling points[28].

In the case of hydrogenation of benzonitrile (BN), the formation of by-product toluene (TOL) is completely undesirable, not only because it lowers the atom economy, but also due to that TOL is carcinogenic[29], which can be fatal for the applications of these contaminated benzylamine (BA) and dibenzylamine (DBA) in pharmaceutical industry. For examples, Pd catalysts often produce BA in majority, but also along with a considerable amount of TOL production[16,19,20]. Pt catalysts show a very different catalytic behavior, by producing DBA as the major product instead, and also along with a few percentage of TOL formation[16,21,22]. Transition metal Ni catalysts are also investigated, but show relatively lower hydrogenation activity even under much higher hydrogen pressures; therein, a mixture of BA, DBA, and the N-benzylidenebenzylamine (DBI) are generally observed[30,31]. The nature of metal catalysts appears to be the crucial factor that governs the reaction selectivity[4,28,32]. Alloying Pd with Ir was shown to be capable of tailoring the product selectivity to a large extend, but still generating an appreciable amount of toxic TOL by-product of ~8%[32]. Incorporation of additives (such as ammonia, NaOH, HCl, acetic acid, $NaH_2PO_4$ et al.) is helpful to prompt primary amine formation[19,28,31,33]. However, such additive process would lead to equipment corrosion issues and raise of the cost for products purification, in addition, the formation of toluene is still inevitable[19]. Recently, hydrogenation of nitriles using a supercritical $CO_2/H_2O$ biphasic solvent and transfer hydrogenation of nitriles with ammonia borane or HCOOH in triethylamine as hydrogen source were reported for selective hydrogenation of nitriles to primary or secondary amines[34–37], while these routes both suffer from high cost due to either the harsh operation pressure or the high cost of the hydrogen source (or solvent). Therefore, it is still an urgent need to develop a heterogeneous catalyst to produce only one of these amines selectively along with complete inhibition of hydrocarbons by-product under a mild and facile reaction condition. Among these protocols, direct hydrogenation of nitriles to secondary amines is particularly desirable, but much more challenging, because of the involvement of complex reaction networks and strong metal-selectivity relations[4,32,38].

Inspired by maximized noble metal utilization in core-shell bimetallic catalysts and the unique coordination and electronic environment in single-atom alloy (SAA) catalysts[39–45], here we report that selective deposition of Pd on silica supported Ni NPs at low coverages using atomic layer deposition (ALD) produces quasi atomically dispersed Pd within the outermost layer of Ni particles to form a core-shell like quasi $Pd_1Ni$ single-atom surface alloy (SASA) structure as confirmed by detailed microscopic and spectroscopic characterization. The resulting isolated Pd atoms and the surrounding Ni atoms act in synergy, break the strong metal-selectivity relations in hydrogenation of BN, and prompt the yield of DBA drastically from ~5 to 97% under mild conditions (80 °C; 0.6 MPa); meanwhile, the carcinogenic TOL by-product is below the detection limit, rendering its practical applications, especially in pharmaceuticals[46], e.g. penicillin[47]. In addition, the activity was also about eight and four times higher than those of monometallic Pd and Pt catalysts, respectively. Theoretical calculations unveil that the strong synergy between isolated Pd atoms and Ni significantly extend the resident time of BI intermediate on $Pd_1Ni$ surface, thus stimulating the exclusive formation of DBA. This method can be extended to a broad scope of nitriles to achieve secondary amines with high yields above 94% under mild conditions, shedding light for controlling the selectivity in hydrogenation of nitriles.

## Results and discussion

**Synthesis and morphology of PdNi bimetallic catalysts**. A set of $PdNi/SiO_2$ bimetallic catalysts with different Pd dispersions were precisely fabricated using a method by combining wet chemistry and ALD (Supplementary Fig. 1). A $Ni/SiO_2$ catalyst with an average particle size of $3.4 \pm 0.7$ nm was first prepared using the deposition-precipitation (DP) method (Supplementary Fig. 2)[48]. After that, with the strategy of low-temperature selective deposition for bimetallic NP synthesis we developed recently[49–51], Pd ALD was executed on the $Ni/SiO_2$ catalyst at 150 °C to deposit Pd selectively on the surface of Ni NPs without any nucleation on $SiO_2$ support (Fig. 1a)[50]. At low Pd coverages, the highly dispersed Pd atoms might become a part of Ni surface lattice to minimize the surface energies (the middle of Fig. 1a). Varying the number of ALD cycles tailors the Pd coverage on Ni NPs precisely. The resulting samples are denoted as $x$Pd-Ni/$SiO_2$, where $x$ represents the number of ALD cycles.

Inductively coupled plasma atomic emission spectroscopy (ICP-AES) analysis (Fig. 1b and Supplementary Table 1) and energy-dispersive spectroscopy (EDS) elemental mapping (Fig. 1c–f) both unambiguously confirmed the selective deposition of Pd on Ni NPs. Figure 1g shows a representative aberration-corrected high-angle annular dark-field scanning transmission electron microscopy (HAADF-STEM) image of 5Pd-Ni/$SiO_2$. Due to the much larger Z value of Pd than Ni, brighter spots highlighted by the yellow arrows in Fig. 1g and Supplementary Fig. 3, along with the intensity profile in Fig. 1h, suggest that Pd atoms were atomically dispersed on the partially oxidized Ni NPs, which might be caused by air exposure during sample transfer. On 20Pd-Ni/$SiO_2$, large Pd ensembles (or even continuous shell) were formed on Ni NPs, as indicated by the brighter shell in Fig. 1i and the line profile analysis in Fig. 1j, k.

**Catalytic performance**. Selective hydrogenation of BN was conducted in a batch reactor at 80 °C using ethanol as the solvent under a $H_2$ pressure of 0.6 MPa. A Pd/$SiO_2$ sample with a Pd particle size of $3.2 \pm 0.3$ nm (Supplementary Fig. 4) was first evaluated. It required ~10 h to complete the reaction (Fig. 2a). The selectivity of BA was about 74% at the earlier stage, but considerably decreased with time. The selectivity of DBA and the undesired hydrogenolysis by-product TOL were about 5 and 21%, respectively, consistent with literature (Supplementary Table 2)[16,19,20]. When conversion was above 90%, the selectivity of TOL increased rapidly to 36% at the expense of the BA, indicating the hydrogenolysis reaction of BA to TOL. Decreasing the reaction temperature from 80 to 60 °C

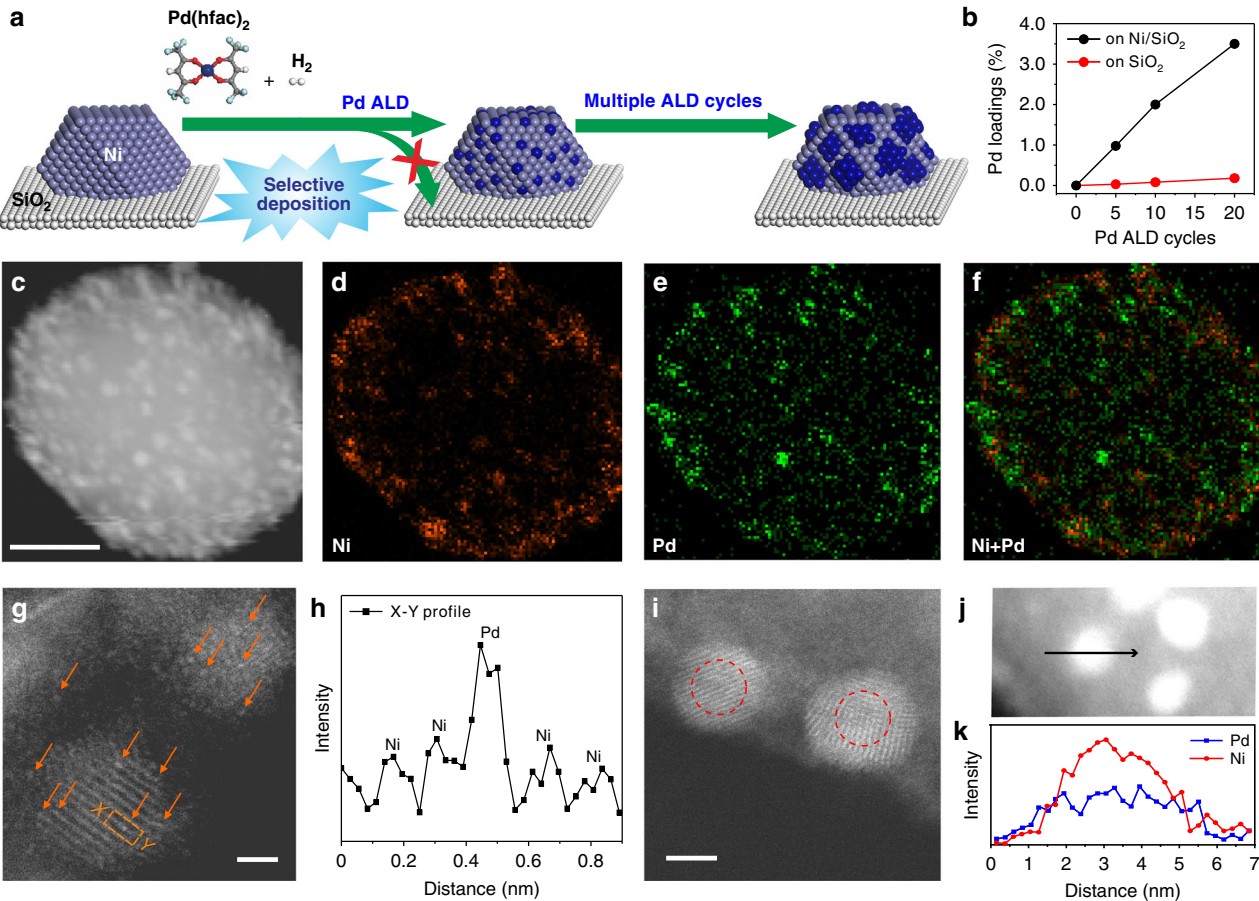

**Fig. 1** Synthesis and morphologies of $x$Pd-Ni/SiO$_2$ bimetallic catalysts. **a** Schematic illustration of synthesis of $x$Pd-Ni/SiO$_2$ bimetallic catalysts using selective Pd ALD. **b** Pd loadings in the $x$Pd-Ni/SiO$_2$ and $x$Pd/SiO$_2$ samples determined by ICP-AES. **c** STEM image of 20Pd-Ni/SiO$_2$ and the corresponding EDS elemental mapping signals: Ni Kα (**d**), Pd Lα (**e**) and the constructed Ni + Pd (**f**). Scale bar in **c** 20 nm. **g** A representative HAADF-STEM image of 5Pd-Ni/SiO$_2$. Isolated Pd single atoms on partially-oxidized Ni NPs were highlighted by brown arrows. Scale bar: 2 nm. **h** Intensity profile along the line X–Y in **g** highlighting the presence of Pd single atoms. **i** A representative HAADF-STEM image of 20Pd-Ni/SiO$_2$. Scale bar: 2 nm. The red circles in **i** highlight contrast of the inner core and outer shell of the NPs. **j** A STEM image of 20Pd-Ni/SiO$_2$. **k** EDS line profile analysis across the NP in **j**

or increasing hydrogenation pressure to 1 MPa did not change the products distribution significantly (Supplementary Table 3). A Pt/SiO$_2$ catalyst (Supplementary Fig. 5), the well-documented catalyst to produce DBA[16,21,22], was also evaluated under the same conditions. It was found that the reaction completed in 9 h (Fig. 2b) with a DBA selectivity of only ~73%. The TOL selectivity was also as high as 11%, consistent with literature (Supplementary Table 2)[16,21,22].

On the 5Pd-Ni/SiO$_2$ sample, the reaction proceeded much quicker, and completed in only ~3 h (Fig. 2c). Very surprisingly, the distribution of products changed completely; therein DBA became the major product, with a selectivity as high as ~97% in the entire range of BN conversions, leading to a high yield of DBA up to 97%. In contrast, BA was decreased sharply to only ~3%, and the undesired hydrogenolysis path to TOL was below the detection limit in the entire range of conversion. More importantly, after the reaction was completed, prolonging the reaction to another 3 h did not alter the DBA selectivity considerably, much better than Pd/SiO$_2$ and Pt/SiO$_2$ (Fig. 2a,b), indication of an effective inhibition of hydrogenolysis of amines to hydrocarbons. Increase of the reaction temperature to 120 °C or the H$_2$ pressure to 1 MPa would reduce the DBA selectivity slightly, while TOL was still effectively prohibited in both cases (Supplementary Table 4). On the other hand, we found that variation of solvents did not change the DBA selectivity considerably, different from the literature[52], although the activity

became slightly lower in non-alcoholic solvents (Supplementary Table 5). This result suggests that the high DBA selectivity achieved on 5Pd-Ni/SiO$_2$ catalyst is solely due to the synergy in Pd$_1$Ni SASA, rather than the solvent effect. These robust catalytic behaviors under different conditions render it convenient for practical operation in a large scale.

Calculations of turnover frequencies (TOFs) reveal that the 5Pd-Ni/SiO$_2$ sample exhibited a highest TOF of 515 h$^{-1}$, which was about eight and four times higher than Pd/SiO$_2$ (64 h$^{-1}$) and Pt/SiO$_2$ (127 h$^{-1}$), respectively, as shown in Fig. 2d. Moreover, the remarkable activity and selectivity achieved on 5Pd-Ni/SiO$_2$ SASA were both much superior than those Pd-, Pt-, Rh-, and Ir-based catalysts reported in literatures (Supplementary Table 2). To note that the conversion of BN was about only 2.6% after 3 h on the Ni/SiO$_2$ sample under the same reaction conditions (Supplementary Fig. 6). Recyclability of 5Pd-Ni/SiO$_2$ was further evaluated, it was found that no significant decay in both selectivity and activity was observed even after the catalyst was recycled for eight times without further calcination/reduction treatments in between (Fig. 2e), indicating the absence of any poisoning or coking. STEM measurements of the recycled sample further confirmed the persistence of the high dispersion of Pd on Ni particles in majority as that in the fresh samples (Supplementary Fig. 7). Accordingly, 5Pd-Ni/SiO$_2$ catalyst was highly active and stable during BN hydrogenation, thereby exhibiting potential practical applications, especially in pharmaceuticals[46], e.g. penicillin[47].

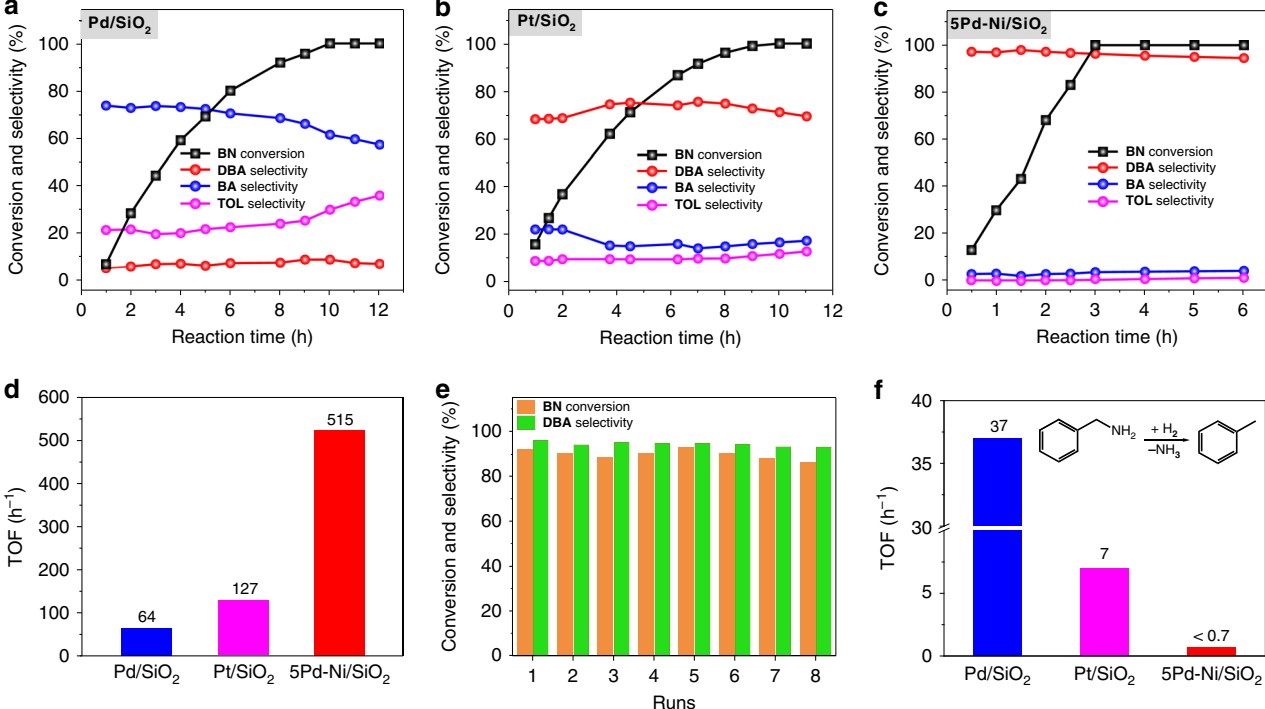

**Fig. 2** Catalytic performances of Pd/SiO$_2$, Pt/SiO$_2$, and 5Pd-Ni/SiO$_2$ catalysts in hydrogenation of BN and hydrogenolysis of BA. Time profiles of hydrogenation of BN over Pd/SiO$_2$ (**a**), Pt/SiO$_2$ (**b**) and 5Pd-Ni/SiO$_2$ (**c**), and their corresponding TOFs (**d**). Reaction conditions: solvent, ethanol, 60 mL; BN, 0.5 g; catalyst, 30 mg; H$_2$ pressure, 0.6 MPa; temperature, 80 °C. **e** Recyclability test of the 5Pd-Ni/SiO$_2$ sample. Reaction conditions: solvent, ethanol, 60 mL; BN, 1 g; catalyst, 100 mg; H$_2$ pressure, 0.6 MPa; temperature, 80 °C; reaction time, 2 h. The larger amount of substrate and catalyst here is only for a purpose of the convenience of recycling. **f** Time profiles of hydrogenolysis of BA over Pd/SiO$_2$, Pt/SiO$_2$ and 5Pd-Ni/SiO$_2$. Reaction conditions: solvent, ethanol, 60 mL; BA, 0.5 g; catalyst, 30 mg; H$_2$ pressure, 0.6 MPa; temperature, 80 °C

Lennon et al. proposed that the TOL formation stems from hydrogenolysis of BA on Pd catalyst, which takes place independently from BN hydrogenation[20]. To get a better understanding of the inhibition of TOL formation on 5Pd-Ni/SiO$_2$ in BN hydrogenation, the BA hydrogenolysis reaction was further performed on these three samples under the same conditions. It was found that BA hydrogenolysis was negligible on 5Pd-Ni/SiO$_2$, but much facile on Pd/SiO$_2$ and Pt/SiO$_2$ (Fig. 2f), unambiguously confirming the effective inhibition of the TOL formation.

Increasing the Pd coverage on Ni decreased the TOF and the yield of DBA considerably by forming more BA product (Supplementary Fig. 8). For instance, on 20Pd-Ni/SiO$_2$, the yield of DBA reduced to 77%, along with a BA yield of 21%. According to the catalytic behavior of Pd/SiO$_2$ (Fig. 2a), the increase of the BA yield on 20Pd-Ni/SiO$_2$ is attributed to the formation of large Pd ensembles at high converges (Fig. 1), providing solid evidence that isolation of Pd with Ni plays the key role for the exclusive DBA formation on 5Pd-Ni/SiO$_2$. However, the TOL formation was still trivial on all PdNi samples (Supplementary Figs. 8 and 9).

Besides above, we found that selective hydrogenation of BN over $x$Pt-Ni/SiO$_2$ ($x = 1$, 3) bimetallic catalysts, synthesized in a similar manner with $x$Pd-Ni/SiO$_2$, also showed remarkable activity improvements and efficient inhibition of TOL formation (Supplementary Table 6). However, over these PtNi bimetallic catalysts, DBI was the major product (>70% selectivity) instead, sharply different from $x$Pd-Ni/SiO$_2$.

**Structural characterization of PdNi bimetallic catalysts.** To establish structure-activity relations, in situ X-ray adsorption fine-structure (XAFS) measurements were first performed on the $x$Pd-

Ni/SiO$_2$ samples ($x = 5$, 10, and 20) at the Pd $K$-edge to investigate the detailed coordination environments of Pd in PdNi bimetallic NPs (Supplementary Figs. 10–12 and Supplementary Note 1). Fourier transforms of the extended X-ray absorption fine structure (EXAFS) spectra of various samples in the real space are shown in Fig. 3a. After in situ H$_2$ reduction at 150 °C, the EXAFS spectrum of 5Pd-Ni/SiO$_2$ exhibited a dominant peak at 2.12 Å, mainly attributed to Pd-Ni coordination[53,54]. EXAFS curve fittings revealed that the Pd-Ni coordination is the dominant one with a coordination number (CN) of 5.5, while the Pd-Pd coordination has a minor contribution with a CN of only 1.2 (Supplementary Figs. 11a and 12a, and Supplementary Table 7), suggesting that Pd atoms are atomically dispersed in majority, in line with the HAADF-STEM observation (Fig. 1g). When the Pd atoms were uniformly distributed over both the surface and the bulk of Ni NPs, it is expected that the Pd-Ni CN will be significantly higher than the average CN for surface atoms[40,55]. In our case, the surface Ni atoms in a 3.4-nm Ni NP have an average Ni–Ni CN of 7.8, according to the cubic-octahedral cluster model[56]. Thus the lower CN of 5.5 for Pd–Ni suggests that the isolated Pd atoms were within the outermost layer of Ni particles in majority to form a core-shell like quasi Pd$_1$Ni SASA structure (the inset of Fig. 3a). To note that this structure with maximized Pd utilization, is sharply different from those SAAs in literature where the isolated secondary metal atoms (B) are rather uniformly dispersed within the primary metal (A) particles with a large CN of A–B bond (often greater than 9)[40,45,55,57].

As increasing the Pd coverage, the peak splits into two peaks at 2.15 and 2.52 Å, assigned to the Pd–Ni and Pd–Pd coordinations, respectively[53,54]. EXAFS curve fittings showed that the CNs of Pd-Ni coordination decreased to 4.3 and 2.7, while the Pd-Pd coordination increased to 3.6 and 6.6 for 10Pd-Ni/SiO$_2$ and

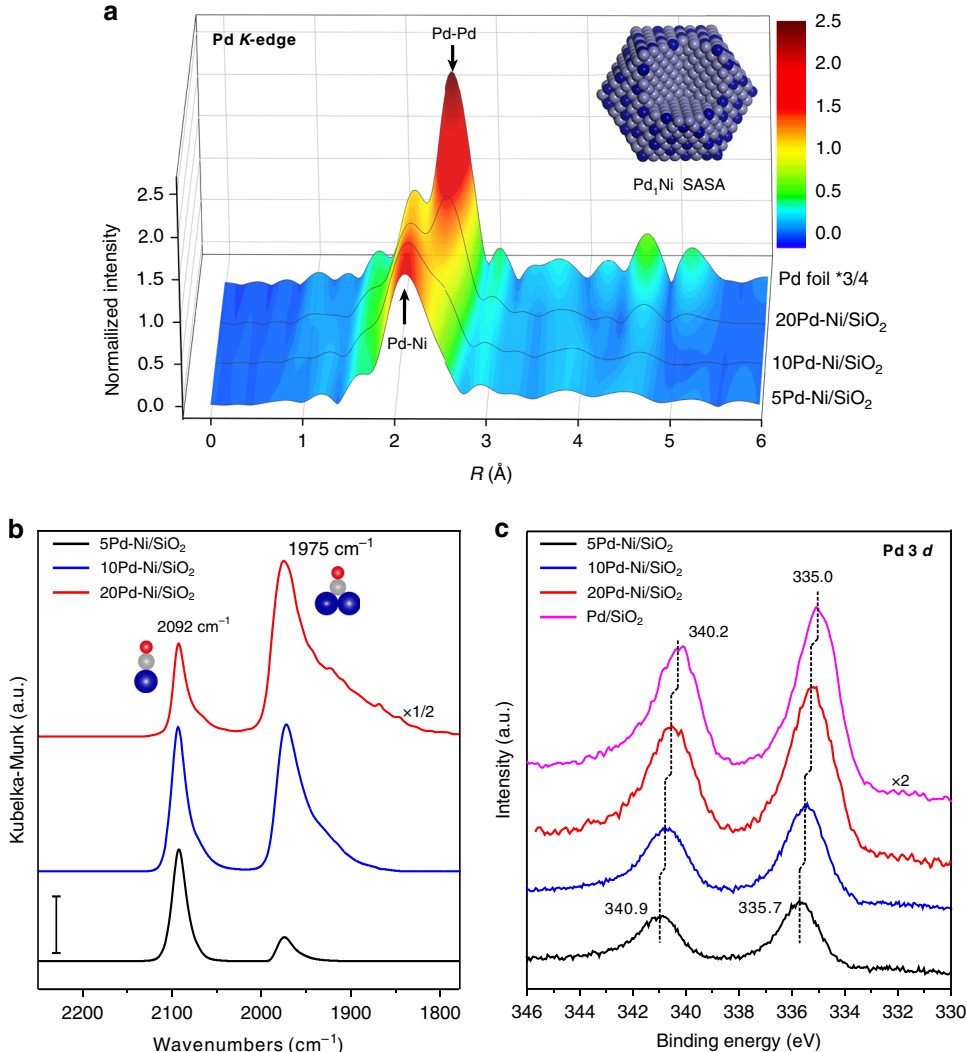

**Fig. 3** Structural characterization of $x$Pd-Ni/SiO$_2$ bimetallic catalysts. **a** In situ Fourier transforms EXAFS spectra of the $x$Pd-Ni/SiO$_2$ samples ($x$ = 5, 10, and 20) and Pd foil reference in the real space at the Pd $K$-edge. A model of core-shell like Pd$_1$Ni SASA structure for 5Pd-Ni/SiO$_2$ is illustrated as the inset in **a**, where the dark blue and light blue balls are Pd and Ni atom, respectively. **b** DRIFTS CO chemisorption of the $x$Pd-Ni/SiO$_2$ samples ($x$ = 5, 10, and 20) at the CO saturation coverage. The dark blue, gray, and red balls in **b** are Pd, C, and O atom, respectively. Scale bar: 0.01. **c** In situ XPS spectra of $x$Pd-Ni/SiO$_2$ samples ($x$ = 5, 10, and 20) and a Pd/SiO$_2$ reference in the Pd $3d$ region

20Pd–Ni/SiO$_2$, respectively (Supplementary Figs. 11b, c, and 12b, c, and Supplementary Table 7). These results describe the evolution of Pd species on Ni from quasi atomically dispersed Pd atoms to large Pd ensembles or even continuous Pd shell, consistent excellently with the STEM observation (Fig. 1).

To further explore the surface structure of PdNi bimetallic NPs, in situ diffuse reflectance infrared Fourier transform spectroscopy (DRIFTS) CO chemisorption measurements were also performed on these samples, since CO is a well-known sensitive probe of Pd ensemble structures[50,58–62]. In this work, after CO exposure, O$_2$ purging was employed to remove chemisorbed CO on Ni, so that the CO on Pd can be illustrated individually (See Supplementary Fig. 13 and Supplementary Note 2 for details). As shown in Fig. 3b, on 5Pd–Ni/SiO$_2$, the dominant peak at 2092 cm$^{-1}$, assigned to the linear CO on Pd, was much stronger than the bridge-bonded CO peak at 1975 cm$^{-1}$, thus again implying that Pd was isolated by the surrounding Ni atoms in majority[50,60,63], which agrees excellently with the EXAFS results in Fig. 3a and the STEM observation (Fig. 1g). As increase of Pd ALD cycles, the bridge-bonded CO peak developed aggressively, clearly demonstrating the formation of large Pd ensembles or continuous Pd shell on the surface of Ni

NPs in these two samples[50]. In situ X-ray photoemission spectroscopy (XPS) measurements in the Pd $3d$ region disclosed a strong electronic interaction between Pd and Ni (Fig. 3c). On 5Pd–Ni/SiO$_2$, we observed a remarkable upwards shift of Pd $3d$ binding energy by 0.7 eV with respect to that of Pd/SiO$_2$, which is attributed to the charge transfer between Pd and Ni[64–66]. The upwards shift gradually became less pronounced as increasing the Pd coverage, implying the surface electronic structure was tending to pure Pd, in line with literature[50,66–68]. In brief, the remarkable activity improvements and selectivity tailoring in Fig. 2 and Supplementary Figs. 8 and 9 are attributed to the ensemble and electronic structure of Pd on Ni.

**Theoretical insight of hydrogenation of BN on metals.** Adamczyk et al. very recently reported detailed calculations of hydrogenation of acetonitrile on Pd and Co catalysts[69–71]. However, to our knowledge, theoretical calculations of hydrogenation of BN on metals have not been reported yet. Here, first-principles DFT calculations on Pd(111) and Pt(111) surfaces were first performed to elucidate the underlying mechanism (Fig. 4 and Supplementary Figs. 14–18). In this work, we only considered the

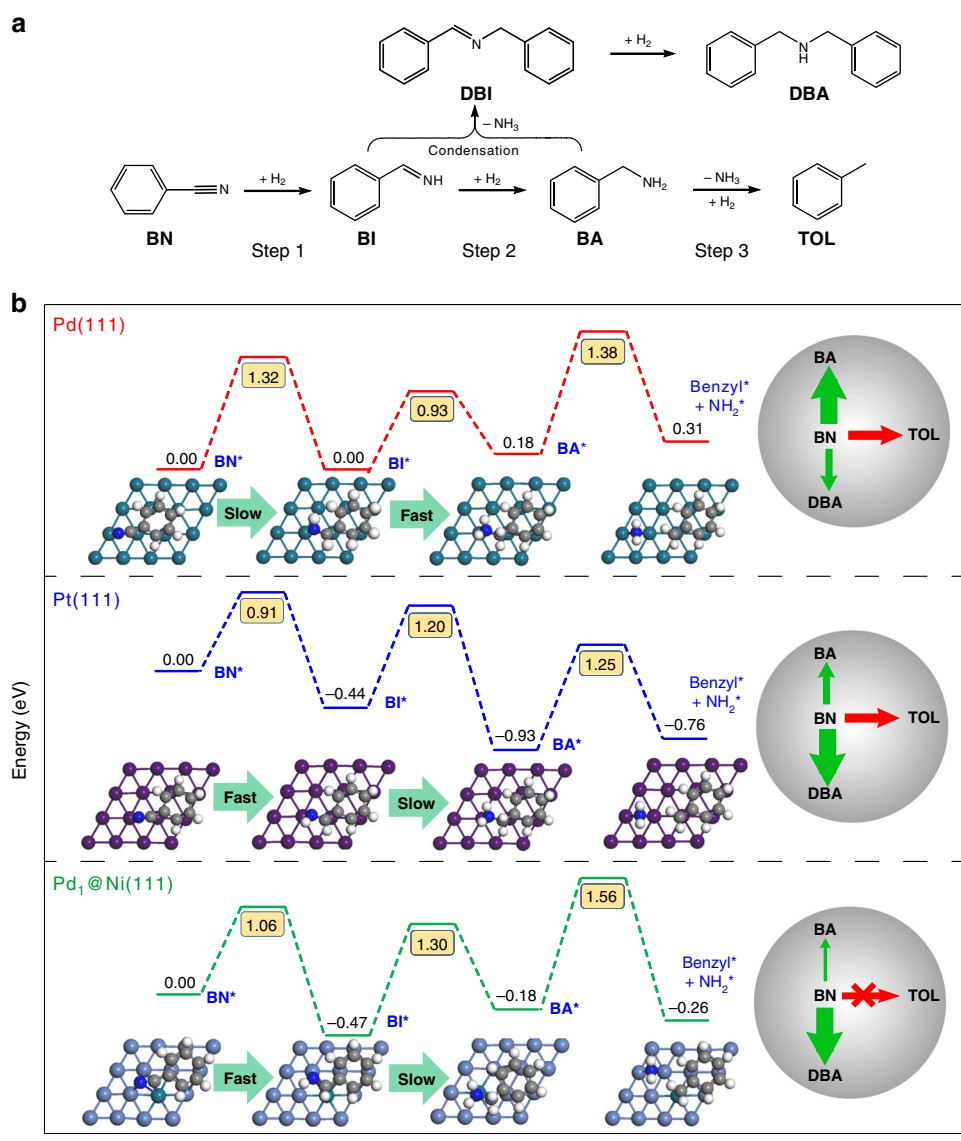

**Fig. 4** The reaction paths of hydrogenation of BN on metal surfaces. **a** Reaction pathways of hydrogenation of BN. **b** Energy profiles of the key intermediates and transition states involved in **a** on Pd(111), Pt(111), and Pd₁@Ni(111) surfaces. The energy barriers of the consecutive hydrogenation and hydrogenolysis steps are highlighted by the yellow squires. The Pd, Pt, Ni, C, N, and H atoms are shown in green, violet, light blue, gray, blue, and white, respectively. On the right side of **b**, the product distribution over the corresponding metal surfaces was highlighted, where the thicker the arrows represents the higher selectivity

reaction pathways of BN hydrogenation to BA and the subsequent BA hydrogenolysis to TOL; while the condensation reaction between BI and BA to DBI, as well as the following hydrogenation of DBI to DBA, were not calculated, because of the limitation of our computing resources for such large molecules (Fig. 4a).

On Pd(111), we found that the first hydrogenation of BN to the benzylideneimine (BI) intermediate, a key intermediate for DBA formation[27], is thermoneutral with an effective barrier of 1.32 eV. The following hydrogenation of BI to BA is slightly endothermic by 0.18 eV, but has a significantly lower effective barrier of 0.93 eV (Fig. 4b, and see Supplementary Figs. 15 and 16 and Supplementary Note 3 for details). On Pt(111), the effective barriers of these two hydrogenation steps show an opposite trend to that on Pd(111): the second hydrogenation step shows a considerably higher effective barrier (1.20 eV) than the first step (0.91 eV) (Fig. 4b, and see Supplementary Figs. 17 and 18 and

Supplementary Note 4 for details). These two consecutive hydrogenation steps are both exothermic by 0.44 and 0.49 eV, respectively. These results infer unambiguously that the considerably higher effective barrier for the second hydrogenation step on Pt(111) favors extending the resident time of the BI surface intermediate, thereby prompting the condensation reaction via nucleophilic attack of BI surface intermediate by BA to form the DBI intermediate (Fig. 4a)[27,32]. Subsequently, the DBI captures surface hydrogen atoms rapidly to produce the desired secondary amine (DBA) product on Pt catalysts[35]. On the contrary, the largely decreased effective barrier for the second hydrogenation step on Pd(111) shortens the resident time of the BI intermediate, and drives the reaction aggressively to the BA formation on Pd catalysts. These calculation results are in excellent consistence with our experimental results in Fig. 2a, b. Therefore, it is concluded that the relative difference of effective energy barriers of first two hydrogenation steps govern the

documented metal-dependent selectivity by regulating the resident time of the BI intermediate. Further hydrogenolysis of BA to the undesired TOL by-product can certainly occur on both Pd(111) and Pt(111), since the corresponding barriers of 1.38 and 1.25 eV are only slightly higher than the barriers of the two hydrogenation steps, which again consists excellently with the experimental results (Fig. 2a, b, f). To our best knowledge, this is the first theoretical view of the metal-selectivity relations on Pd and Pt surfaces in hydrogenation of BN.

In sharp contrast, on Pd$_1$@Ni(111) SASA, where the isolated Pd atoms are within the outmost layer of Ni lattices according to EXAFS analysis (Fig. 3a), the reaction profile changes dramatically (Fig. 4b and see Supplementary Figs. 19 and 20 and Supplementary Note 5 for detials): the hydrogenation of BN to the BI intermediate becomes more facile on Pd$_1$@Ni(111) (barrier 1.06 eV, exothermic 0.47 eV) than on Pd(111) (barrier 1.32 eV, exothermic 0 eV). On the contrary, the second hydrogenation step becomes more difficult with a considerably higher effective barrier of 1.30 eV. According to the knowledge learned on Pd (111) and Pt(111), facilitation of the first hydrogenation step but suppression of the second one would drastically improve the DBA formation, consistent with our experimental observation (Fig. 2c). Meanwhile, it is also found that the BI intermediate adsorbs considerably stronger on Pd$_1$@Ni(111) (2.74 eV) than on Pd(111) (2.59 eV) and Pt(111) (2.08 eV) (Supplementary Fig. 21). Such stronger adsorption would reduce the mobility of BI on Pd$_1$@Ni (111) and further accelerate the condensation reaction. These findings provide a molecular-level insight of breaking the metal-dependent selectivity in BN hydrogenation over Pd$_1$Ni SASA. Besides above, benefiting from the surrounding Ni (Supplementary Figs. 21–23 and Supplementary Note 6), hydrogenolysis of BA on Pd$_1$@Ni(111) has a much higher barrier of 1.56 eV than that on Pd(111) (1.38 eV) (Fig. 4b), again elucidating the effective inhibition of undesired hydrogenolysis of BA to TOL on 5Pd-Ni/SiO$_2$ (Fig. 2c, f). The electronic perturbation of Pd from the underlying Ni as suggested by XPS might play the important role (Fig. 3c).

However, the DFT results seem not elucidate the activity enhancement on Pd$_1$Ni SASA on its own, since the highest effective energy barrier of BN hydrogenation to BA on Pd$_1$@Ni (111) surface (1.30 eV) is very close to that on Pd(111) (1.32 eV) and Pt(111) surface (1.20 eV). In fact, it is well-known that the hydrogenation activity not only depends on energetic profiles, but also correlates strongly with the competitive adsorption of substrate molecule and H$_2$[16]. For bimetallic catalysts, hydrogen spillover could play very important roles for the activity enhancement[39,72]. For example, in hydrogenation of 3-nitrostyrene, Peng et al. reported that Pt$_1$Ni SAA catalyst had a TOF of ~1800 h$^{-1}$, much higher than that of Pt single atoms supported on active carbon, TiO$_2$, SiO$_2$, and ZSM-5. The remarkable activity of Pt$_1$/Ni nanocrystals was attributed to sufficient hydrogen supply because of spontaneous dissociation of H$_2$ on both Pt and Ni atoms as well as facile diffusion of H atoms on Pt$_1$/Ni nanocrystals[72]. In our work, we speculate that spontaneous dissociation of H$_2$ on both Pd and Ni atoms provides a reservoir of active H atoms on Pd$_1$Ni SASA surface, thus greatly accelerating the sequential hydrogenations.

**Substrate exploration.** Finally, we further evaluated the 5Pd-Ni/SiO$_2$ SASA sample in hydrogenation of a broad scope of nitrile substrates. As shown in Fig. 5, hydrogenation of aromatic nitriles with either electron-donating groups (R = CH$_3$, OCH$_3$) or electron-withdrawing groups (R = F, CF$_3$) and aliphatic nitriles, proceeded smoothly under mild conditions without any additives and leading to excellent yield of the corresponding secondary

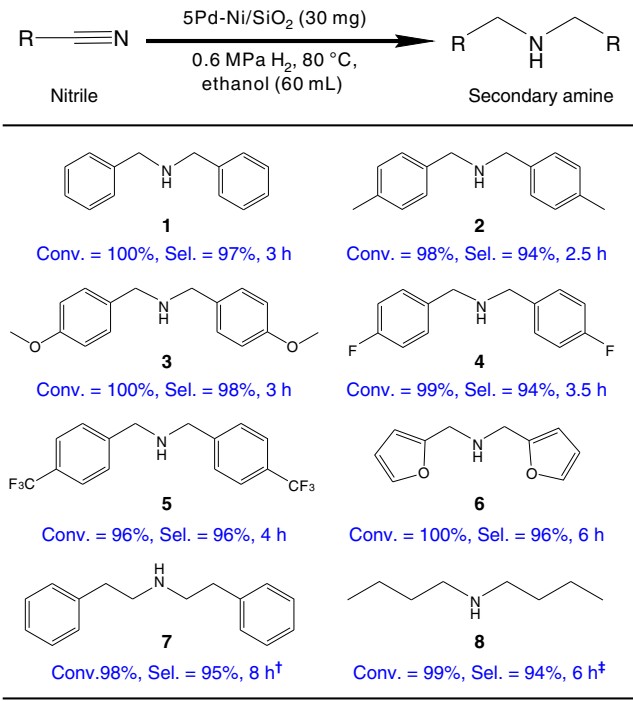

**Fig. 5** Catalytic performance of 5Pd-Ni/SiO$_2$ catalyst in hydrogenation of substituted nitriles. Reaction conditions: nitrile, 0.5 g; ethanol, 60 mL; catalyst, 30 mg; temperature, 80 °C; H$_2$, 0.6 MPa. †0.25 g nitrile, 65 °C. ‡0.125 g nitrile, 65 °C

amines above 94%. The major by-products were the corresponding primary amines, while the hydrogenolysis by-products were completely inhibited in all cases, demonstrating the great potentials for industrial applications.

In summary, we have demonstrated that isolating Pd with Ni breaks the strong metal-selectivity relations in hydrogenation of nitriles, and prompts the yield of secondary amines drastically up to >94% over a broad scope of nitriles under middle conditions. Meanwhile, the activity was also remarkably enhanced to about eight and four times higher than those of Pd and Pt catalysts, respectively. Importantly, the resulting materials also showed excellent recyclability and a complete inhibition of the formation of hydrogenolysis by-product, demonstrating the great potentials for practical applications. DFT calculations, to our best knowledge, provided the first theoretical view of the metal-selectivity relations on Pd and Pt surfaces, and unveiled the synergy in Pd$_1$Ni SASA for the switch of reaction pathway from primary amines on Pd to the exclusive formation of secondary amines. It should be noted that selective Pd ALD is essential to achieve the high DBA yield with complete TOL inhibition, while such selective deposition might not be applicable to other supports such as Al$_2$O$_3$. Nonetheless, these findings open a promising avenue to rational design of metal catalyst with controlled selectivity and activity.

### Data availability

The data underlying Figs. 1–5, Supplementary Figs. 2, 4–13, 16, 18, 20, and 23, DFT calculated XYZ coordination parameters of reactants, intermediates, products, and transition states on Pd(111), Pt(111), and Pd$_1$@Ni(111) surfaces are provided as a Source Data file. The other data that support the findings of this study are available from the corresponding author upon request.

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

## Acknowledgements

This work was supported by the National Natural Science Foundation of China (Grants 21673215, 21688102, and 21703222), the National Key Research & Development Program of China (Grant 2016YFA02006), the Fundamental Research Funds for the Central Universities (WK2060030029), Users with Excellence Program of Hefei Science Center CAS (2019HSC-UE016), and the Max-Planck Partner Group. The calculations were performed on the supercomputing system in the Supercomputing Center of University of Science and Technology of China and the High-performance Computing Platform of Anhui University. The authors also gratefully thank the BL14W1 beamline at the Shanghai Synchrotron Radiation Facility (SSRF), the BL10B beamline at National Synchrotron Radiation Laboratory (NSRL), China.

## Author contributions

J.L. designed the experiments; H.W. synthesized and characterized the catalysts and performed the catalytic performance evaluation; H.W., S.W., W.L., and Z.S. performed the XAFS measurements; H.W. X.Z., H.P., and J.Z. did the XPS characterization; Y.L. did the STEM measurements; Q.G. helped catalyst synthesis; Q.L. and J.Y. did the DFT calculations; J.L. and H.W. co-wrote the manuscript, and all the authors contributed to the overall scientific interpretation and edited the manuscript.

## Competing interests

The authors declare no competing interests.
