## [Peer Review File · Nature Communications]

Reviewers' comments:

Reviewer #1 (Remarks to the Author):

Dear Members of the Editorial Board,

This paper by Prof. Lu and co-workers reports a very interesting and thought provoking study of quasi Pd1Ni single-atom surface alloys to enable hydrogenation of nitriles to secondary amines. The authors have synthesized and characterized the PdNi bimetallic catalysts, tested the catalytic performance of these catalysts by hydrogenating benzonitrile (BN) and other related nitriles, and performed theoretical calculations by way of Density Functional Theory (DFT).

The catalysts were prepared by wet chemistry and Atomic Layer Deposition (ALD), and the morphology was characterized by inductively coupled plasma atomic emission spectroscopy (ICP-AES), energy dispersive spectroscopy (EDS), transmission electron microscopy (TEM), aberration-corrected high angle annular dark-field scanning transmission electron microscopy (HAADF-STEM). Additionally, catalyst structural characterization was performed using in situ X-ray absorption fine-structure measurements (XAFS) including Fourier transforms of the extended X-ray absorption fine structure (EXAFS). Also, in situ diffuse reflectance infrared Fourier transform spectroscopy (DRIFTS) carbon monoxide chemisorption and in situ X-ray photoemission spectroscopy (XPS) measurements were made to further probe catalyst structure. Lastly, in situ X-ray absorption near edge structure (XANES) measurements were also performed to probe catalyst structure.

Catalytic performance was assessed in batch reactor experiments, and products were identified using gas chromatography (GC) and then compared to literature experimental data. The computational tool for the theoretical DFT calculations was the Vienna Ab initio Simulation Package (VASP) software which is widely used in both industry and academia. This package was then used to study the energetics behind the reaction mechanism on Pd(111), Pt(111), and Pd1@Ni(111) surfaces.

The systematic and nearly comprehensive approach to catalyst design described above was followed for careful synthesis, testing, and analysis of a core-shell like Pd1Ni single-atom surface alloy (SASA) catalyst. As the present work extends the understanding of the rational catalyst design for nitrile hydrogenation process with claims that are relatively well supported, convincing, and likely of interest to a wide community, I believe that the paper could be published in Nature Communications. However, several revisions are needed in both the main body text and supporting information (SI) regarding clarity, claims, and originality:

1. Authors have employed a large range scientific techniques to support the claims of the catalyst presented. It would be very helpful for the reader, especially one not familiar with heterogeneous catalysis, if one figure can be created to show the entire process of rational catalyst design from start to finish and how all the methods support the claims of the catalyst.
2. What are the specific properties of this catalyst that can be improved further; in other words, what are the drawbacks of this catalyst compared to commercially used catalysts? One sentence or more should be added to the main text or SI.
3. When the authors mention 'cancerogenous' with respect to toluene, do the authors instead mean 'carcinogenic'? Please change or clarify in the abstract and main text (pages 2 and 3).
4. On page 5 of the main body text, the authors should clarify that the full hydrogenation to the secondary amine was not presented with the theoretical DFT calculations. Please see comments 11 and 12 below regarding such work from our research group which should be cited as prior art.
5. On page 6 for Figure 1, it unclear how the graphics in Figure 1a relate to the catalyst which is the

subject of this paper. The authors should clarify if the Pd is nucleating on the Ni surface, or if the Pd is becoming part of the Ni lattice as it is modelled in the DFT calculations (see bottom of Figure 4b). One or more sentences should be added to the main text or in the Figure 1 caption.

6. For Figure 2, it would be helpful to explain in the main text or SI why there are different feed to catalyst weight ratios for the Pt, Pd, and Pd-Ni catalysts hydrogenation of BN, recyclability, and hydrogenolysis of benzyl amine (BA). One or more sentences should be added to the main text or SI to clarify whether the same catalyst weight and feed nitrile weight were used in each study of hydrogenation, recyclability, and hydrogenolysis.

7. Was coking or catalyst poisoning observed, if not, would the authors expect to see this phenomena at longer reaction times or harsher reactor conditions? One or more sentences should be added to the main text or SI.

8. The authors have chosen ethanol as a solvent for the reactor experiments. One or more sentences should be added to clarify whether ethanol plays a role in the reaction pathways, and how the findings may change with another solvent.

9. On page 8 line 170, I believe that there is a typo as Figure 2b references Pt not Pd.

10. Also, on page 8 lines 168-171, the authors should explain in more detail why the selectivity is not altered in 3hrs for the Pd-Ni catalyst, and how this conclusion may change at longer reactor times. One sentence here to clarify would be helpful in the main text.

11. On page 12 lines 277-281, the conclusions of the theoretical calculations is based on analysis which does not cover the complete reaction network to secondary amine. This missing aspect of the analysis should be highlighted in the main text or SI with one or more sentences, as the claims drawn are affected by this missing analysis. Also, see comment 12 below which may be referenced in your analysis.

12. In the literature review or analysis of DFT results, it is asked that the authors cite relevant prior art from our research group where we performed detailed periodic plane wave DFT calculations and reactor modeling on Pd and related catalysts for nitrile hydrogenation, as this work was recently published by our group for model nitrile species and extended to more complex nitriles like the one in this current study under review:

- o A. J. Adamczyk, First-principles analysis of acetonitrile reaction pathways to primary, secondary, and tertiary amines on Pd(111), *Surface Science* 682 (2019) 84-98.
- o G. Lozano-Blanco, A. J. Adamczyk, Cobalt-catalyzed nitrile hydrogenation: Insights into the reaction mechanism and product selectivity from DFT analysis, *Surface Science* 688 (2019) 31-44.
- o G. Lozano-Blanco, B. J. Tatarchuk, A. J. Adamczyk, Building a microkinetic model from first principles for higher amine synthesis on Pd catalyst, *ACS Industrial & Engineering Chemistry Research* (2019) Accepted.

13. In Figure 4b, the bottom potential energy surface for the Pd-Ni catalyst is not referenced in the text. One or more sentences should be added to the main text.

14. The authors should provide the XYZ coordinates (reactants, intermediate, products, and transition states on Pd, Pt, and Pd-Ni surfaces) for all of the DFT calculations to allow for reproducibility of the results presented. This information can be added the SI.

15. The authors compare three catalysts in figure 4; however, a Ni only catalyst on a support is not presented and quite important for comparison. The authors should present such results if they have it available, or mention in the text accordingly. That is, how significantly different is the Pd-Ni catalyst

from a Ni only supported catalyst. This part is missing from the study and very important.

16. Supplementary figure 8 is not referenced in the text, and there is no explanation of the in situ XANES method in the text. The authors should add a paragraph describing their methodology and reference the figure in the SI text.

17. The authors did not perform in situ X-ray powder diffraction (XRD) to also probe the catalyst structure and complement the other characterization methods performed. Could the authors clarify why this method was not performed for the reader in the SI text?

18. Supplementary figure 10 is not referenced in the text. The authors should reference the figure in the SI text.

19. On page 22/39 of SI, 'disassociation' may be more correctly written as 'dissociation'.

20. Supplementary figures 17, 18, and 19 are not referenced in the text. The authors should reference each of these figures in the SI text.

The authors are encouraged to reach out to our research group if more detailed multi-scale modeling studies are desired to continue their rational catalyst design efforts. Our current nitrile hydrogenation models are available to be tuned to their experimental data, if desired, and these validated process models can undoubtedly provide more atomistic-level insight into the detailed kinetics and resultant selectivity observed in their own studies.

Excellent work! It was a pleasure to review this study.

Sincerely,

Andrew J. Adamczyk, PhD

Assistant Professor of Chemical Engineering
Auburn University
Auburn, Alabama, USA
Email: aja0056@auburn.edu

Reviewer #2 (Remarks to the Author):

This manuscript describes the preparation and characterization of quasi Pd1Ni single-atom surface alloy (SASA) structure onto SiO₂. The materials have been used for hydrogenation of nitriles with the excellent yield of dibenzylamine.

Although this topic is important, and the findings are potentially interesting, quite some work is needed to improve the manuscript. A few areas where the presentation is not clear and others where more detailed discussion is necessary. The authors must address the points below before publication. So this manuscript can be accepted for publication in Nature Communication after minor revisions.

1. It is inaccurate that the authors' conversion of secondary amines increased from 5% to 97%. The reaction conditions are dependence in the different catalytic systems. The conditions of 0.6 Mpa and 80 oC, may not be the optimum condition of Pd/SiO₂ for the selective hydrogenation of nitriles to obtain secondary amines. The yield of BA maybe lower and the yield of DBA maybe increase when the temperature is lowered. Therefore, the author should explore the selectivity of Pd/SiO₂ for DBA under different reaction conditions, and obtain optimal conditions to compare with Pd1Ni/SiO₂.

2. Why does the authors choose 0.6 MPa, 80 oC as the optimal reaction condition? The authors should supplement the experiment on the exploration of reaction conditions.

3. Since toluene is only produced at high temperatures, the authors can continue to increase the reaction temperature to 100 or 120 °C to investigate whether the reaction will also produce toluene to verify the suitability of the catalyst.

4. Does the Pd on the catalyst agglomerate after the reaction? At least, the recycled catalyst must be characterized by TEM.

5. The Pt/SiO₂ catalyst is the well-documented catalyst to produce DBA. Why did the author not use Pt to prepare the catalyst? The authors should prepare PtNi/SiO₂ catalysts by the same atomic layer deposition method to compare with the Pd-based catalysts in the paper to highlight the performance excellence of the Pd-based catalyst.

6. In the hydrogenation of nitriles, the concentration of the reactants and the kind of the solvent affect the reaction yield. The authors can refer to this related literature of Yingwei Li's group (DOI 10.1021/acscatal.8b01834). And the authors should add the experiment about the reaction of different solvent quantities and solvent types.

7. The product of Fig. 5(4) is not bis(4-fluorobenzyl)amine. Since fluorine is easily detached, has defluorination occurred in this reaction?

8. Some closely related literatures should be cited and discussed in the introduction part, e.g. Moderate Activity from Trace Palladium Alloyed with Copper for the Chemoselective Hydrogenation of -CN and -NO₂ with HCOOH DOI: 10.1002/slct.201902057; A ppm level Rh-based composite as an ecofriendly catalyst for transfer hydrogenation of nitriles: triple guarantee of selectivity for primary amines DOI: 10.1039/c8gc03595d.

Reviewer #3 (Remarks to the Author):

This work presents the synthesis of single atom surface alloy (SASA) of Pd₁Ni with excellent catalytic performance in nitriles hydrogenation to secondary amines with high activity and selectivity. The catalysts were synthesized with ALD approach and characterized in details via STEM, XAS, XPS and DRIFTS. The catalytic results were correlated with the DFT insights. I found this a very interesting work and well executed. I recommend its acceptance after a few points to be addressed.

- There was no description of the DFT result on the Pt@Ni(111) surface. The authors must have omitted this, they need to add this part into the manuscript.
- Although the DFT calculation results can help to understand the selectivity to DBA, it is not clear why the TOF for SASA sample is significantly higher than the individual Pt or Pd catalysts. The authors should elaborate this clearly.
- Some proofreading is useful for the manuscript. For example, Page 9 Line 211: "are atomically" should be "are atomic"; Page 12 Line 285-286: the sentence should be changed to: "These calculation results are in excellent consistence with xxx"

Response to reviewer #1:

General comment: This paper by Prof. Lu and co-workers reports a very interesting and thought provoking study of quasi Pd₁Ni single-atom surface alloys to enable hydrogenation of nitriles to secondary amines. The authors have synthesized and characterized the PdNi bimetallic catalysts, tested the catalytic performance of these catalysts by hydrogenating benzonitrile (BN) and other related nitriles, and performed theoretical calculations by way of Density Functional Theory (DFT).

The catalysts were prepared by wet chemistry and Atomic Layer Deposition (ALD), and the morphology was characterized by inductively coupled plasma atomic emission spectroscopy (ICP-AES), energy dispersive spectroscopy (EDS), transmission electron microscopy (TEM), aberration-corrected high angle annular dark-field scanning transmission electron microscopy (HAADF-STEM). Additionally, catalyst structural characterization was performed using in situ X-ray absorption fine-structure measurements (XAFS) including Fourier transforms of the extended X-ray absorption fine structure (EXAFS). Also, in situ diffuse reflectance infrared Fourier transform spectroscopy (DRIFTS) carbon monoxide chemisorption and in situ X-ray photoemission spectroscopy (XPS) measurements were made to further probe catalyst structure. Lastly, in situ X-ray absorption near edge structure (XANES) measurements were also performed to probe catalyst structure.

Catalytic performance was assessed in batch reactor experiments, and products were identified using gas chromatography (GC) and then compared to literature experimental data. The computational tool for the theoretical DFT calculations was the Vienna Ab initio Simulation Package (VASP) software which is widely used in both industry and academia. This package was then used to study the energetics behind the reaction mechanism on Pd(111), Pt(111), and Pd₁@Ni(111) surfaces.

The systematic and nearly comprehensive approach to catalyst design described above was followed for careful synthesis, testing, and analysis of a core-shell like Pd₁Ni single-atom surface alloy (SASA) catalyst. As the present work extends the understanding of the rational catalyst design for nitrile hydrogenation process with claims that are relatively well supported, convincing, and likely of interest to a wide community, I believe that the paper could be published in Nature Communications. However, several revisions are needed in both the main body text and supporting information (SI) regarding clarity, claims, and originality:

Response: We gratefully thank Prof. Adamczyk for his careful review and positive comments on our study.

Comment 1: Authors have employed a large range scientific techniques to support the claims of the catalyst presented. It would be very helpful for the reader, especially one not familiar

with heterogeneous catalysis, if one figure can be created to show the entire process of rational catalyst design from start to finish and how all the methods support the claims of the catalyst.

Response: We gratefully thank Prof. Adamczyk's constructive suggestion.

According to the reviewer's suggestion, we drew a schematic figure to illustrate the entire process of catalyst synthesis (**Fig. R1**). **Figure R1** was added into the revised Supplementary Information (SI) as **Supplementary Figure 1**.

Figure R1 | Schematic illustration of synthesis of $x\text{Pd-Ni/SiO}_2$ bimetallic catalysts by combining wet-chemistry with selective Pd ALD.

Comment 2: What are the specific properties of this catalyst that can be improved further; in other words, what are the drawbacks of this catalyst compared to commercially used catalysts? One sentence or more should be added to the main text or SI.

Response: We gratefully thank Prof. Adamczyk's valuable comment.

One drawback of this catalyst is from the selective Pd ALD. In our strategy, we

demonstrated that selective deposition of Pd is essential to achieve the high DBA selectivity with complete TOL inhibition. In the synthesis of 5Pd/Ni/SiO₂ catalyst, Pd can selectively deposit on Ni particles, but not on the SiO₂ support, while this might not be applicable to other supports, such as alumina. The ALD conditions for achieving selective Pd ALD over other usual-used supports require further exploration.

In our revised manuscript, we included the above discussion of the limitations of selective ALD in the Conclusion section on page 17, highlighted in yellow.

Comment 3: When the authors mention ‘cancerogenous’ with respect to toluene, do the authors instead mean ‘carcinogenic’? Please change or clarify in the abstract and main text (pages 2 and 3).

Response: We have replaced the word ‘*cancerogenous*’ with ‘*carcinogenic*’ through the entire manuscript as your suggestion, which were highlighted in yellow.

Comment 4: On page 5 of the main body text, the authors should clarify that the full hydrogenation to the secondary amine was not presented with the theoretical DFT calculations. Please see comments 11 and 12 below regarding such work from our research group which should be cited as prior art.

Response: We gratefully thank Prof. Adamczyk’s valuable comments.

As shown in **Fig. 4a**, the pathway to dibenzylamine (DBA) formation involves condensation of benzylideneimine (BI) and benzylamine (BA) to form N-benzylidenebenzylamine (DBI) and then further hydrogenation of DBI to DBA. Modeling the large molecules of DBI and DBA requires huge unit cells, which is a great challenge for us due to the limitation of computing resources.

In our revised manuscript, we have clarified it on page 12, highlighted in yellow. In addition, the excellent references provided by the reviewer were also cited, which are highlighted in yellow.

Comment 5: On page 6 for Figure 1, it unclear how the graphics in Figure 1a relate to the catalyst which is the subject of this paper. The authors should clarify if the Pd is nucleating on the Ni surface, or if the Pd is becoming part of the Ni lattice as it is modelled in the DFT calculations (see bottom of Figure 4b). One or more sentences should be added to the main text or in the Figure 1 caption.

Response-2: We gratefully thank Prof. Adamczyk’s valuable concerns and suggestions.

Figure 1a provides a schematic illustration of selective deposition of Pd on Ni nanoparticles to synthesize xPd-Ni/SiO₂ catalyst, which is the central part of the catalyst synthesis. The Pd coverage is precisely tailored as increasing the number of ALD cycles.

According to our EXAFS results (**Fig. 3a** and **Supplementary Table 7**). We found that the Pd-Ni CNs decreased from 5.5 to 4.3 and to 2.7, along with the increase of Pd-Pd CNs from 1.2 to 3.6 and to 6.6, respectively for the samples of 5Pd-Ni/SiO₂, 10Pd-Ni/SiO₂, and 20Pd-Ni/SiO₂. The large Pd-Ni CNs of 5.5 for 5Pd-Ni/SiO₂, indicates that the Pd is within the Ni lattice, which might be due to the strong Pd-Ni interactions. As increasing the Pd coverage, the Pd-Ni CNs decreased quickly to 2.7, implying that the Pd-Pd interactions might be stronger than Pd-Ni, leading to the Pd islands formation as shown in **Fig. 1a**.

We have added additional clarifications on pages 5 in our revised manuscript, which are highlighted in yellow.

Comment 6: For Figure 2, it would be helpful to explain in the main text or SI why there are different feed to catalyst weight ratios for the Pt, Pd, and Pd-Ni catalysts hydrogenation of BN, recyclability, and hydrogenolysis of benzyl amine (BA). One or more sentences should be added to the main text or SI to clarify whether the same catalyst weight and feed nitrile weight were used in each study of hydrogenation, recyclability, and hydrogenolysis.

Response: We gratefully thank Prof. Adamczyk's valuable concerns and suggestions.

We confirmed that each study of hydrogenation (**Fig. 2a-c**) and hydrogenolysis (**Fig. 2f**), the ratios of the benzonitrile (BN) feed (0.5 g) to catalyst weight (30 mg) were actually same. While for the recyclability test, we increased both the benzonitrile feed (1.0 g) and the catalyst weight (100 mg). This is only for a purpose of the convenience of catalyst recovering in a large number of recycling.

We have clarified it in the caption of Fig. 2e on page 8 in our revised manuscript, which is highlighted in yellow.

Comment 7: Was coking or catalyst poisoning observed, if not, would the authors expect to see this phenomena at longer reaction times or harsher reactor conditions? One or more sentences should be added to the main text or SI.

Response: We gratefully thank Prof. Adamczyk's valuable comments.

During the entire recyclability test, the 5Pd-Ni/SiO₂ catalyst was collected by simple centrifugations and then drying at 50 °C for 1 h after each run. The collected catalyst was then directly used for the next run without any further calcination/reduction treatments. Given the unchanged selectivity and activity in 8 runs, we believed there was no coking or poisoning phenomena during the reaction in our mild reaction conditions (80 °C, 0.6 MPa). We expect that both coking and poisoning would be negligible even under harsher reaction conditions because of the high stability of benzyl ring in BN molecule.

We have added the above discussion on page 9 in our revised manuscript, which is highlighted in yellow.

Comment 8: The authors have chosen ethanol as a solvent for the reactor experiments. One or more sentences should be added to clarify whether ethanol plays a role in the reaction pathways, and how the findings may change with another solvent.

Response: We gratefully thank Prof. Adamczyk's valuable comments.

To evaluate the solvent effect, we further performed the BN hydrogenation reaction over 5Pd-Ni/SiO₂ in different solvents. As shown in **Table R1**, this catalyst exhibited remarkably high selectivity (> 92%) to DBA along with the absence of toluene formation in all solvents, further confirming that the high DBA selectivity achieved on 5Pd-Ni/SiO₂ catalyst is solely due to the synergy in Pd₁Ni SASA, rather than the solvent effect.

Nonetheless, we did find that the activity of the 5Pd-Ni/SiO₂ catalyst in BN hydrogenation with alcoholic solvents was considerably higher than those with non-alcoholic solvents. This might be attributed to the higher solubility of hydrogen in such polar solvents (*Winterbottom, J. et al., J. Catal. 1976, 44, 271-280*).

We added the above discussion into the revised manuscript on page 8. **Table R1** was also included in the revised SI as **Supplementary Table 5**. All changes were highlighted in yellow.

Table R1 | Catalytic performance of the 5Pd-Ni/SiO₂ catalyst in BN hydrogenation with different solvents.

Entry	Solvent	Reaction time (h)	Conversion (%)	Selectivity (%)			TOF (h ⁻¹)
				TOL	BA	DBA	
1	ethanol	1	29.5	N.D.	2.8	97.2	515
		3	99.5	N.D.	3.5	96.5	
2	methanol	1	28.0	N.D.	6.4	93.6	490
		3	95.2	N.D.	7.4	92.6	
3	isopropanol	1	15.3	trace (<0.6)	2.4	97.0	134
		9	97.1	trace (<0.7)	4.4	94.9	
4	n-hexane	1	6.9	N.D.	0.5	99.5	60
		12	67.8	N.D.	1.0	99.0	
5	dichloromethane	1	2.8	trace (<0.6)	0.6	98.8	24
		12	26.3	trace (<0.9)	1.2	97.9	

Reaction conditions: Solvent, 60 mL; BN, 0.5 g; catalyst, 30 mg; H₂ pressure, 0.6 MPa; temperature, 80 °C. TOFs were evaluated after proceeding the reaction for 1 h.

N.D. means "non-detected".

Comment 9: On page 8 line 170, I believe that there is a typo as Figure 2b references Pt not Pd.

Response: We gratefully thank Prof. Adamczyk for pointing out this typo, we have corrected it accordingly.

Comment 10: Also, on page 8 lines 168-171, the authors should explain in more detail why the selectivity is not altered in 3hrs for the Pd-Ni catalyst, and how this conclusion may change at longer reactor times. One sentence here to clarify would be helpful in the main text.

Response: We gratefully thank Prof. Adamczyk's comments.

After the reaction completed, all BN was transferred into DBA. Prolonging the reaction for another 3 h did not change the selectivity of DBA. This is because that hydrogenolysis of amine is completely prohibited on 5Pd-Ni/SiO₂. This unique property was confirmed by the control experiment of BA hydrogenolysis reaction, where 5Pd-Ni/SiO₂ showed a significantly lower activity compared with Pd/SiO₂ and Pt/SiO₂ catalysts (**Fig. 2f**)

We added the above discussion into the revised manuscript on page 7 in our revised manuscript, highlighted in yellow.

Comment 11: On page 12 lines 277-281, the conclusions of the theoretical calculations is based on analysis which does not cover the complete reaction network to secondary amine. This missing aspect of the analysis should be highlighted in the main text or SI with one or more sentences, as the claims drawn are affected by this missing analysis. Also, see comment 12 below which may be referenced in your analysis.

Response: We gratefully thank Prof. Adamczyk's valuable comments.

The complete reaction network to secondary amine involves the condensation of BI with BA to DBI, followed by DBI hydrogenation (**Fig. 4a**). We didn't apply theoretical calculations to this part is mainly because that the spontaneously modeling the energetics of two large molecules (BI + BA), DBI intermediate and the DBA product requires huge unit cells and is very challenging for us due to the limitation of computing resources.

Nonetheless, we found that the relative effective energy barriers for the two hydrogenation steps (BN to BI and BI to BA) determine the resident time of BI intermediate on catalyst surfaces, which further governing the condensation reaction between BI and BA then the DBA formation. The above findings elucidate excellently with the experimental results on Pd, Pt and Pd₁Ni SASA catalysts, thus are unambiguously reliable.

In our revised manuscript, we have clarified it on page 12, highlighted in yellow. In addition, the excellent references provided by the reviewer were also cited, which are highlighted in yellow.

Comment 12: In the literature review or analysis of DFT results, it is asked that the authors cite relevant prior art from our research group where we performed detailed periodic plane wave DFT calculations and reactor modeling on Pd and related catalysts for nitrile hydrogenation, as this work was recently published by our group for model nitrile species and extended to more complex nitriles like the one in this current study under review:

> o A. J. Adamczyk, First-principles analysis of acetonitrile reaction pathways to primary, secondary, and tertiary amines on Pd(111), Surface Science 682 (2019) 84-98.

> o G. Lozano-Blanco, A. J. Adamczyk, Cobalt-catalyzed nitrile hydrogenation: Insights into the reaction mechanism and product selectivity from DFT analysis, Surface Science 688 (2019) 31-44.

> o G. Lozano-Blanco, B. J. Tatarchuk, A. J. Adamczyk, Building a microkinetic model from first principles for higher amine synthesis on Pd catalyst, ACS Industrial & Engineering Chemistry Research (2019) Accepted.

Response: We gratefully thank Prof. Adamczyk for providing us the wonderful references. Unfortunately we can't find the third one, we only found the first two references online.

These excellent references provided by the reviewer have been cited on page 12 in our revised manuscript, which are highlighted in yellow.

Comment 13: In Figure 4b, the bottom potential energy surface for the Pd-Ni catalyst is not referenced in the text. One or more sentences should be added to the main text.

Response: We gratefully thank Prof. Adamczyk's insightful suggestion.

We should apologize for our mistake that the discussion of the DFT results on the Pd₁@Ni(111) surface was somehow omitted when we submitted the manuscript. We have added this part on pages 14-15 in our revised manuscript, highlighted in yellow.

Comment 14: The authors should provide the XYZ coordinates (reactants, intermediate, products, and transition states on Pd, Pt, and Pd-Ni surfaces) for all of the DFT calculations to allow for reproducibility of the results presented. This information can be added the SI.

Response: We gratefully thank Prof. Adamczyk's insightful suggestion.

According to the reviewer's suggestion, we have provided the XYZ coordinates of the DFT calculations on Pd, Pt and Pd₁@Ni surface. However, we didn't include it in the SI because of the huge data (147 pages). We uploaded it separately as the online source data instead.

Comment 15: The authors compare three catalysts in figure 4; however, a Ni only catalyst on a support is not presented and quite important for comparison. The authors should present such results if they have it available, or mention in the text accordingly. That is, how significantly different is the Pd-Ni catalyst from a Ni only supported catalyst. This part is missing from the study and very important.

Response: We gratefully thank Prof. Adamczyk's constructive suggestion.

According to the reviewer's kind suggestion, we have performed DFT calculations of BN hydrogenation on Ni(111) surface. Similar to those on Pd(111), Pt(111) and Pd₁@Ni(111), the reaction pathways involved in BN hydrogenation to BA and subsequently hydrogenolysis to TOL were considered.

Figure R2 | Energy profiles of the benzyl nitrile hydrogenation and hydrogenolysis paths on Ni(111).

As shown in **Fig. R2**, the effective energy barriers of the first two hydrogenation steps (BN hydrogenation to BI and then BI hydrogenation to BA) are very close on Ni(111), 1.22 eV and 1.21 eV, respectively. Therefore, the coupling reaction of BI with BA can be highly possible on Ni(111). In addition, we found that the BI intermediate adsorbs considerably stronger on Ni(111) (3.03 eV) than on Pd(111) (2.59 eV) and Pt(111) (2.08 eV) (**Fig. R3**). Such stronger adsorption would reduce the mobility of BI on Ni(111), further favoring the coupling reaction.

Besides above, **it is very interesting that hydrogenolysis of BA on Ni(111) has a much higher barrier of 1.71 eV, which would effectively inhibit the hydrogenolysis reaction on Ni(111) (Fig. R2). This value follows the trend that the presence of Ni elevates the barrier of hydrogenolysis considerably from 1.38 eV on Pd(111) to 1.56 eV on Pd₁@Ni(111).**

It should be noted that although the highest effective energy barrier among the two hydrogenation steps on Ni(111) (1.22 eV) is slightly lower than that on Pd(111) (1.32 eV, see **Fig. 4b**) and Pt(111) (1.20 eV, see **Fig. 4b**), the activity of Ni/SiO₂ catalyst in BN hydrogenation is unexpectedly lower than that of either Pd/SiO₂ or Pt/SiO₂ (**Supplementary Fig. 6**). One possible reason is due to that the adsorptions of BN and its derived intermediates are too strong on Ni(111) (**Fig. R3**).

We have added the Figs. R2 and R3 to our revised SI as Supplementary Figs. 23 and 21, respectively. The optimized geometries of BN, its derived key intermediates and transition states on Ni(111) were also shown in Supplementary Fig. 22.

Meanwhile, the above discussion was also added to our revised SI on page 33 for a better understanding in detail. All changes were highlighted in yellow.

Figure R3 | Adsorption energies of the BN, BA and the BI intermediate on Pd(111), Pt(111), Pd₁@Ni(111) and Ni(111) surfaces.

Comment 16: Supplementary figure 8 is not referenced in the text, and there is no explanation of the in situ XANES method in the text. The authors should add a paragraph describing their methodology and reference the figure in the SI text.

Response: We gratefully thank Prof. Adamczyk's insightful suggestion.

Supplementary Figure 8 (renumbered to **Supplementary Figure 10** in the revised SI) shows the X-ray adsorption near-edge structure (XANES) spectra of 5Pd-Ni/SiO₂, 10Pd-Ni/SiO₂ and 20Pd-Ni/SiO₂ and the reference Pd foil for comparison.

To keep the length of main manuscript short, a supplementary text for describing this Figure was added in SI. We have also referenced this figure in our revised manuscript on page 10, highlighted in yellow.

Comment 17: The authors did not perform in situ X-ray powder diffraction (XRD) to also probe the catalyst structure and complement the other characterization methods performed. Could the authors clarify why this method was not performed for the reader in the SI text?

Response: We gratefully thank Prof. Adamczyk's insightful comment.

We do believe that the in situ XRD would be a powerful technique to characterize the structure evolution. However, one major reason why we did not use it in this study is that the Ni particle size is rather small (average in 3.4 nm), the XRD pattern of Ni/SiO₂ only exhibits weak reflection peaks as shown in **Fig. R4**. For the pattern of 5Pd-Ni/SiO₂ sample, we can't observe any visible difference from that of Ni/SiO₂ sample. This result implies that PdNi does

not form a uniform alloy, which consists very well with the EXAFS data where the Pd atoms only locates at the surface of Ni particles to form the Pd₁Ni SASA structure.

However, considering its poor data quality, we would not include it in our manuscript.

Figure R4 | Ex situ XRD patterns of Ni/SiO₂ and 5Pd-Ni/SiO₂ catalysts.

Comment 18: Supplementary figure 10 is not referenced in the text. The authors should reference the figure in the SI text.

Response: We gratefully thank Prof. Adamczyk’s careful review. This figure is renumbered as **Supplementary Figure 12** in our revised SI. **We have referenced it in our revised manuscript on page 10, highlighted in yellow.**

Comment 19: On page 22/39 of SI, ‘disassociation’ may be more correctly written as ‘dissociation’.

Response: We gratefully thank Prof. Adamczyk’s careful review. We have changed “*disassociation*” to “*dissociation*” in our revised SI accordingly, which was highlighted in yellow.

Comment 20: Supplementary figures 17, 18, and 19 are not referenced in the text. The authors should reference each of these figures in the SI text.

Response: We gratefully thank Prof. Adamczyk’s careful review and insightful suggestion.

Supplementary Figure 17-19 in our former SI are the detailed information about DFT calculations of BN hydrogenation on Pd₁@Ni(111) surface. **We should apologize for our mistake that the discussion of the above process was somehow omitted when we submitted the manuscript. Now we have added this missing part on pages 14-15 in our revised manuscript. Meanwhile, these figures have been renumbered to Supplementary Figure 19-21 in our revised SI and referenced in the discussion part mentioned above. All changes were highlighted in yellow.**

Response to reviewer #2:

General comment: This manuscript describes the preparation and characterization of quasi Pd₁Ni single-atom surface alloy (SASA) structure onto SiO₂. The materials have been used for hydrogenation of nitriles with the excellent yield of dibenzylamine. Although this topic is important, and the findings are potentially interesting, quite some work is needed to improve the manuscript. A few areas where the presentation is not clear and others where more detailed discussion is necessary. The authors must address the points below before publication. So this manuscript can be accepted for publication in Nature Communication after minor revisions.

Response: We gratefully thank the reviewer for his/her careful review and positive comments on our study.

Comment 1: It is inaccurate that the authors' conversion of secondary amines increased from 5% to 97%. The reaction conditions are dependence in the different catalytic systems. The conditions of 0.6 MPa and 80 °C, may not be the optimum condition of Pd/SiO₂ for the selective hydrogenation of nitriles to obtain secondary amines. The yield of BA maybe lower and the yield of DBA maybe increase when the temperature is lowered. Therefore, the author should explore the selectivity of Pd/SiO₂ for DBA under different reaction conditions, and obtain optimal conditions to compare with Pd₁Ni/SiO₂.

Response: We gratefully thank the reviewer's valuable comments and insightful suggestions.

We agree with the reviewer that the catalytic performance depends on the reaction conditions. **To avoid any misunderstandings from readers, we further clarify our descriptions in both abstract and introduction from "under mild conditions." to "under mild conditions (80 °C; 0.6 MPa)" on pages 2 and 5 in our revised manuscript, highlighted in yellow.**

According to the reviewer's excellent suggestion, here we further evaluated the Pd/SiO₂ catalyst under different reaction conditions. The results are shown in **Table R2**. We found that the selectivity of DBA slightly increased from 8.7% to 11.2% at the later stage of the reactions, as decreasing the reaction temperature from 80 to 60 °C (see the entries 1-3). The selectivity of BA also showed a similar trend with that of DBA, which is mainly due to the significantly suppressed TOL formation under lower temperature (25.2% at 80 °C vs 13.2% at 60 °C).

Comparison of entry 1 with entry 3 showed that when the reaction temperature decreased from 80 to 60 °C, the activity decreased by 75%, while the DBA selectivity only increased slightly from 8.7% to 11.2% at ~ 93% BN conversion. Moreover, the TOL byproduct still formed in a considerable amount (13.2%). **Apparently, such optimization of reaction conditions would not be able to produce DBA selectively without any carcinogenic TOL formation as what we achieved on Pd₁Ni/SiO₂.** The reaction conditions (80 °C; 0.6 MPa) we used in our manuscript are the optimized conditions when activity and selectivity are both

considered.

We have added the above discussion on page 7 in our revised manuscript, highlighted in yellow. Table R2 was also added to our revised SI as Supplementary Table 3. All changes were highlighted in yellow.

Table R2 | Catalytic performance of the Pd/SiO₂ catalyst in BN hydrogenation with different reaction conditions

Entry	Temperature (°C)	Reaction time (h)	Conversion (%)	Selectivity (%)			TOF (h ⁻¹) ^d
				TOL	BA	DBA	
1 ^a	80	2	28.3	21.2	73.7	5.1	64
		9	95.6	25.2	66	8.7	
2 ^a	70	2	13.0	15.5	76.7	7.8	31
		18	92.0	22.1	69.8	8.1	
3 ^b	60	2	13.4	12.3	78.3	9.4	15.9
		20	93.0	13.2	75.6	11.2	
4 ^c	80	2	23.0	20.2	75.2	4.6	54
		13	97.0	23.8	72.4	3.8	

^a Reaction conditions: Solvent, 60 mL ethanol; BN, 0.5 g; catalyst, 30 mg; H₂ pressure, 0.6 MPa;

^b Reaction conditions: Solvent, 60 mL ethanol; BN, 0.5 g; catalyst, 60 mg; H₂ pressure, 0.6 MPa;

^c Reaction conditions: Solvent, 20 mL ethanol; BN, 0.5 g; catalyst, 30 mg; H₂ pressure, 0.6 MPa;

^d TOFs were evaluated after proceeding the reaction for 2 h.

Comment 2: Why does the authors choose 0.6 MPa, 80 °C as the optimal reaction condition? The authors should supplement the experiment on the exploration of reaction conditions.

Response: We gratefully thank the reviewer's valuable comments and insightful suggestions.

As we all know, developing catalytic hydrogenation reaction under lower temperature and hydrogen pressure is crucial for cost saving in industrial application. We chose 0.6 MPa, 80 °C for catalytic performance evaluation of our Pd₁Ni SASA catalyst because it is a quite milder reaction condition both in temperature and hydrogen pressure compared with other catalytic processes reported in previous literature (**Supplementary Table 1**).

Here we further tested the catalyst under harsher reaction conditions (1.0 MPa or 100, 120 °C), the results were shown in **Table R3**. Increasing reaction temperature or H₂ pressure would significantly enhance the activity in BN hydrogenation reaction, whereas the selectivity to DBA would decrease slightly. For example, the DBA selectivity dropped to 92% and 80%, as increasing the H₂ pressure to 1.0 MPa or rising the reaction temperature to 120 °C, respectively. **Interestingly, the formation of TOL was still nearly inhibited, indicating the**

robust applications. Therefore, the reaction condition (0.6 MPa, 80 °C) that we chose is a proper condition with sufficient activity and high selectivity.

We have added the above discussion on page 7 in our revised manuscript, highlighted in yellow. Table R3 was also added to our revised SI as Supplementary Table 4. All changes were highlighted in yellow.

Table R3 | Catalytic performance of 5Pd-Ni/SiO₂ catalyst in BN hydrogenation with different reaction conditions

Entry	Temp. (°C)	H ₂ pressure (MPa)	Reaction time (h)	Conversion (%)	Selectivity (%)			TOF (h ⁻¹)
					TOL	BA	DBA	
1	80	0.6	1	29.5	N.D.	2.8	97.2	515
			3	99.5	N.D.	3.5	96.5	
2	80	1.0	1	36.0	N.D.	5.3	94.7	630
			2.5	97.9	N.D.	7.9	92.1	
3	100	0.6	1	53.5	N.D.	11.8	88.1	935
			2	98.0	trace (<0.5)	13.5	86.0	
4	120	0.6	0.5	60.3	trace(<0.3)	15.9	83.8	2109
			1	99.3	trace (<0.7)	19.3	80.0	

Reaction conditions: Solvent, 60 mL ethanol; BN, 0.5 g; catalyst, 30 mg. TOFs were evaluated after proceeding the reaction for 1 h.

N.D. means “non-detected”.

Comment 3: Since toluene is only produced at high temperatures, the authors can continue to increase the reaction temperature to 100 or 120 °C to investigate whether the reaction will also produce toluene to verify the suitability of the catalyst.

Response: We gratefully thank the reviewer’s valuable comments and insightful suggestions.

Please see our detailed discussion in our response to the Comment 2 above.

Comment 4: Does the Pd on the catalyst agglomerate after the reaction? At least, the recycled catalyst must be characterized by TEM.

Response: We gratefully thank the reviewer’s valuable suggestions.

We have performed additional HAADF-STEM characterization on the 8-cycle-used 5Pd-Ni/SiO₂ sample. As shown in **Fig. R5**, we found that the Pd atoms were still atomically-dispersed on Ni NPs in majority (highlighted by the brown arrows), implying that there were no visible aggregation of Pd after reaction. On the other hand, the selectivity to DBA remained constant during 8-cycle recycling tests, providing additional strong evidence that the Pd did not

aggregate during reactions, since the presence of large Pd ensembles will induce considerable DBA selective decrease (Supplementary Fig. 8).

We have added the above discussion on page 9 in our revised manuscript. Figure R5 was also added to our revised SI as Supplementary Fig. 7. All changes were highlighted in yellow.

Figure R5 | Representative HAADF-STEM images of the 5Pd-Ni/SiO₂ catalyst after 8-cycle reaction. **a**, A low-magnification STEM image. **b**, The corresponding particle size distribution. **c-i**, High-magnification STEM images at different locations. Isolated Pd single atoms on partially-oxidized Ni NPs are highlighted by brown arrows. In addition, small Pd ensembles (highlighted by brown circles in **h** and **i**) were also observed which is consistent with the DRIFTS CO chemisorption and XAFS results.

Comment 5: The Pt/SiO₂ catalyst is the well-documented catalyst to produce DBA. Why did the author not use Pt to prepare the catalyst? The authors should prepare PtNi/SiO₂ catalysts by the same atomic layer deposition method to compare with the Pd-based catalysts in the paper to highlight the performance excellence of the Pd-based catalyst.

Response: We gratefully thank the reviewer's valuable comments and suggestions.

In this work, our goal is to provide an atomic-level understanding of the metal-selectivity relations (BA forms dominantly on Pd catalysts, while DBA forms majorly on Pt catalysts), and to design an advanced catalyst to produce DBA selectively without any formation of toluene (TOL). The huge change in DBA selectivity from ~5% on Pd/SiO₂ to 97% on Pd₁Ni SASA

(80 °C and 0.6 MPa) manifests our advanced strategy for catalyst design.

Evaluating the PtNi/SiO₂ catalyst is an excellent suggestion. According to this suggestion, we prepared two PtNi/SiO₂ catalysts using a similar procedure as that used for PdNi/SiO₂. Therein, 1 and 3 cycles of Pt ALD were performed on the Ni/SiO₂ sample at 150 °C using trimethyl(methylcyclopentadienyl)platinum(IV) (MeCpPtMe₃) and O₂, which were denoted as 1Pt-Ni/SiO₂ and 3Pt-Ni/SiO₂, respectively.

Table R4 | Catalytic performance of 1Pt-Ni/SiO₂, 3Pt-Ni/SiO₂ and Pt/SiO₂ catalysts in BN hydrogenation.

Entry	Catalyst	Reaction time (h)	Conversion (%)	Selectivity (%)				TOF (h ⁻¹)
				TOL	BA	DBI	DBA	
1	1Pt-Ni/SiO ₂	1	14.0	N.D.	7.8	89.7	2.5	1110
		6	95.2	N.D.	9.1	77.6	13.3	
2	3Pt-Ni/SiO ₂	1	15.1	N.D.	0.4	94.5	5.1	517
		6	98.1	N.D.	15.5	70.3	14.2	
3	Pt/SiO ₂	1	16.0	9.2	22.3	N.D.	68.5	127
		9	99	11.1	16.1	N.D.	72.8	

Reaction conditions: Solvent, 60 mL ethanol; BN, 0.5 g; catalyst, 30 mg, reaction temperature, 80 °C, H₂ pressure, 0.6 MPa. TOFs were evaluated after proceeding the reaction for 1 h. N.D. means “non-detected”.

In BN hydrogenation, we found the activities of 1Pt-Ni/SiO₂ and 3Pt-Ni/SiO₂ were 1110 and 517 h⁻¹, respectively, again much higher than that of Pt/SiO₂ catalyst (127 h⁻¹) (Table R4), in line with the activity improvement achieved on 5Pd-Ni/SiO₂ (Fig. 2). In addition, we also found that TOL was again not formed on 1Pt-Ni/SiO₂ and 3Pt-Ni/SiO₂, in line with that on 5Pd-Ni/SiO₂.

However, we found that the major product on 1Pt-Ni/SiO₂ and 3Pt-Ni/SiO₂ was N-benzylidenebenzylamine (DBI) with a selectivity >70% at high conversions, sharply different from the dominant DBA formation on Pt/SiO₂ (entry 3). This result agrees well with the previous work about PtNi bimetallic catalyst for nitrile hydrogenation reported by Prof. Yingwei Li’s group, where the DBI selectivity was about >99% (Long, J. et al., *AIChE J.* 2014, 60, 3565-3576). **The large selectivity difference between PdNi and PtNi suggests that the Pd₁Ni SASA catalyst is the only proper candidate for active and selective DBA production without any TOL formation.**

The above discussion of PtNi catalyst was added in the revised manuscript on page 9. We also added Table R4 into the revised SI as Supplementary Table 6. The synthesis

procedure of PtNi/SiO₂ catalysts were also described in our revised SI. All changes were highlighted in yellow.

Comment 6: In the hydrogenation of nitriles, the concentration of the reactants and the kind of the solvent affect the reaction yield. The authors can refer to this related literatures of Yingwei Li's group (DOI 10.1021/acscatal.8b01834). And the authors should add the experiment about the reaction of different solvent quantities and solvent types.

Response: We gratefully thank the reviewer for his/her insightful suggestions and providing us the valuable reference.

According to the reviewer's kind suggestion, we evaluated the 5Pd-Ni/SiO₂ catalyst with five difference solvents in BN hydrogenation (three polar solvents: ethanol, methanol and isopropanol; two apolar solvents: n-hexane and dichloromethane). **We found that the solvents had a very limited impact on selectivity, where the selectivity of DBA were above 93% at near 100% BN conversion in all kind of solvents (Table R1). These results suggest the high DBA selectivity achieved on 5Pd-Ni/SiO₂ catalyst is solely due to the PdNi synergy effect rather than the solvent effect.**

However, we did find that the activity varies considerably with solvents. As shown in **Table R1**, the activity is higher in polar, alcoholic solvents rather than that in apolar, non-alcoholic solvents, among which ethanol is the best solvent for this PdNi catalyst. Detailed discussion can be seen in our response to the **comment 8 of reviewer #1**.

To evaluate the effect of solvent quantities, we also performed additional experiments. As shown by the entries 1 and 4 in **Table R2**, we found that the selectivity remained unchanged when we reduced the volume of ethanol solvent form 60 mL to 20 mL, while the activity was only slightly decreased from 64 to 54 h⁻¹ on Pd/SiO₂ catalyst. Apparently, the solvent quantity again has very limited impact on the catalytic performance.

We added the above discussion in the revised manuscript on page 8. We also added Tables R1 and R2 into our revised SI as Supplementary Tables 5 and 3, respectively. All changes are highlighted in yellow.

Comment 7: The product of Fig.5(4) is not bis(4-fluorobenzyl)amine. Since fluorine is easily detached, has defluorination occurred in this reaction?

Response: We gratefully thank the reviewer for his/her careful review and apologize for the mistake.

The product in **Fig. 5(4)** is revised to bis(4-fluorobenzyl)amine, as shown in **Fig. R6**. Meanwhile, we have updated the **Fig. 5** in our revised manuscript on **page 16**.

In this reaction, the selectivity of bis(4-fluorobenzyl)amine is up to 94% at near 100% BN conversion, the only by-product that we detected is 4-fluorobenzylamine. No visible defluorination occurred in this reaction under such mild conditions (80 °C, 0.6 MPa H₂).

Figure R6 | Catalytic performance of 5Pd-Ni/SiO₂ Catalyst in hydrogenation of substituted nitriles. Reaction conditions: nitrile, 0.5 g; ethanol, 60 mL; catalyst, 30 mg; temperature, 80 °C; H₂, 0.6 MPa. [†]0.25 g nitrile, 65 °C. [‡]0.125 g nitrile, 65 °C.

Comment 8: Some closely related literatures should be cited and discussed in the introduction part, e.g. Moderate Activity from Trace Palladium Alloyed with Copper for the Chemoselective Hydrogenation of -CN and -NO₂ with HCOOH DOI: 10.1002/slct.201902057; A ppm level Rh-based composite as an ecofriendly catalyst for transfer hydrogenation of nitriles: triple guarantee of selectivity for primary amines DOI: 10.1039/c8gc03595d.

Response: We gratefully thank the reviewer for his/her valuable suggestions and providing us wonderful references.

We have referenced these two articles in the introduction of our revised manuscript on page 4, highlighted in yellow.

Response to reviewer #3:

General comment: This work presents the synthesis of single atom surface alloy (SASA) of Pd₁Ni with excellent catalytic performance in nitriles hydrogenation to secondary amines with high activity and selectivity. The catalysts were synthesized with ALD approach and characterized in details via STEM, XAS, XPS and DRIFTS. The catalytic results were correlated with the DFT insights. I found this a very interesting work and well executed. I recommend its acceptance after a few points to be addressed.

Response: We gratefully thank the reviewer's positive comments on our manuscript.

Comment 1: There was no description of the DFT result on the Pt@Ni(111) surface. The authors must have omitted this, they need to add this part into the manuscript.

Response: We gratefully thank the reviewer's careful review and valuable comments.

We should apologize for our mistake of omitting the description part of DFT result on the Pd₁@Ni(111) surface somehow during submission (we believe that the reviewer referred to Pd₁Ni(111), instead of Pt₁Ni(111)).

We have added the discussion theoretical results on Pd₁@Ni(111) on pages 14-15 in our revised manuscript, highlighted in yellow.

Comment 2: Although the DFT calculation results can help to understand the selectivity to DBA, it is not clear why the TOF for SASA sample is significantly higher than the individual Pt or Pd catalysts. The authors should elaborate this clearly.

Response: We gratefully thank the reviewer's valuable comments and insightful suggestions.

In this work, DFT calculations, to our best knowledge, provided the first theoretical view of the metal-selectivity relations on Pd and Pt surfaces, and unveiled the synergy in Pd₁Ni SASA for the switch of reaction pathway from primary amines on Pd to the exclusive formation of secondary amines.

According to the catalyst activity, our DFT results showed that the energy barriers of the rate-determining step were 1.32, 1.20 and 1.30 eV on Pd(111), Pt(111) and Pd₁@Ni(111), respectively. Clearly, the energy barriers themselves does not explain the much higher activity achieved on the 5Pd-Ni/SiO₂ SASA catalyst. In fact, it is well known that the hydrogenation activity not only depends on energetic profiles, but also correlates strongly with the competitive adsorption of substrate molecule and H₂ (Bakker, W. *et al.*, *J. Catal.* **2010**, 274, 176-191). For bimetallic catalysts, hydrogen spillover can also play significant roles for the activity enhancement (Kyriakou, G. *et al.*, *Science* **2012**, 335, 1209-1212). For example, in hydrogenation of 3-nitrostyrene, Peng *et al.* reported a highly active Pt₁Ni SAA catalyst with a TOF of ~1800 h⁻¹, much higher than that of Pt single atoms supported on active carbon, TiO₂, SiO₂, and ZSM-5. The mechanistic studies reveal that the remarkable activity of Pt₁/Ni

nanocrystals derived from sufficient hydrogen supply because of spontaneous dissociation of H₂ on both Pt and Ni atoms as well as facile diffusion of H atoms on Pt₁/Ni nanocrystals (Peng, Y. et al., *Nano Lett.* **2018**, *18*, 3785-3791). Their results are shown in **Fig. R7** for your convenience.

According to above discussion, the activity enhancement on Pd₁Ni SASA catalyst in BN hydrogenation might be also possible due to the optimal competitive adsorption of BN with H₂ on active sites. Spontaneous dissociation of H₂ on both Pd and Ni atoms in Pd₁Ni SASA provides a reservoir of active H atoms on surface, thus accelerating the sequential hydrogenation.

We added the above discussion into our revised manuscript on pages 15-16, the references mentioned here was also cited. All changes are highlighted in yellow.

Figure R7 | Schematic illustration of H₂ dissociation and H diffusion on Pt₁/Ni(111) and activity comparison of different Pt₁ and Pt NPs catalysts in hydrogenation of 3-nitrostyrene. Reproduced from the literature (Peng, Y. et al., *Nano Lett.* **2018**, *18*, 3785-3791).

Comment 3: Some proofreading is useful for the manuscript. For example, Page 9 Line 211: "are atomically" should be "are atomic"; Page 12 Line 285-286: the sentence should be changed to: "These calculation results are in excellent consistence with xxx"

Response: We gratefully thank the reviewer for his/her careful review and valuable comments. We have changed them accordingly in our revised manuscript, highlighted in yellow.

REVIEWERS' COMMENTS:

Reviewer #1 (Remarks to the Author):

Dear Members of the Editorial Board,

I thank the authors for addressing my comments. The authors have sufficiently addressed my comments; however, the paper that the authors could not find is listed here with DOI:

G. Lozano-Blanco, B.J. Tatarchuk, A.J. Adamczyk, Building a microkinetic model from first principles for higher amine synthesis on Pd catalyst, ACS Industrial & Engineering Chemistry Research (2019). <https://pubs.acs.org/doi/abs/10.1021/acs.iecr.9b03577>

If the authors could please cite this paper as well, the manuscript will have sufficiently captured the prior art.

Overall, this paper by Prof. Lu and co-workers reports a very thorough study of quasi Pd1Ni single-atom surface alloys to enable hydrogenation of nitriles to secondary amines and deserves to be published.

Sincerely,
Andrew J. Adamczyk, PhD

Assistant Professor of Chemical Engineering
Auburn University
Auburn, Alabama, USA
Email: aja0056@auburn.edu

Reviewer #2 (Remarks to the Author):

This manuscript describes an interesting and novel study of quasi Pd1Ni single-atom surface alloy catalyst to enable hydrogenation of nitriles to secondary amines without toluene. The catalysts were prepared by wet chemistry and Atomic Layer Deposition (ALD), and characterized by TEM, HAADF-STEM, XAS, XPS and DRIFTS. The authors performed theoretical calculations by way of DFT. After the revision, the experimental data of this manuscript is more sufficient and convincing. As this study open a new way to design metal catalysts in selective hydrogenation of nitrile, I believe this manuscript can be accepted for publication in Nature Communication.

Reviewer #3 (Remarks to the Author):

The authors have addressed my previous comments and I recommend acceptance of the manuscript for publication in its current form.

We greatly all the reviewers for giving us the very positive comments and constructive suggestions, which have made the manuscript to be significantly strengthened and much better presented.

REVIEWERS' COMMENTS:

Reviewer #1 (Remarks to the Author):

Dear Members of the Editorial Board,

I thank the authors for addressing my comments. The authors have sufficiently addressed my comments; however, the paper that the authors could not find is listed here with DOI: G. Lozano-Blanco, B.J. Tatarchuk, A.J. Adamczyk, Building a microkinetic model from first principles for higher amine synthesis on Pd catalyst, ACS Industrial & Engineering Chemistry Research (2019). <https://pubs.acs.org/doi/abs/10.1021/acs.iecr.9b03577>

Response: We gratefully thank Prof. Adamczyk for proving us the link of this excellent reference! We have added it into our revised manuscript.

Overall, this paper by Prof. Lu and co-workers reports a very thorough study of quasi Pd1Ni single-atom surface alloys to enable hydrogenation of nitriles to secondary amines and deserves to be published.

Response: We gratefully thank Prof. Adamczyk for his careful review and positive comments on our study.

Sincerely,

Andrew J. Adamczyk, PhD

Assistant Professor of Chemical Engineering

Auburn University

Auburn, Alabama, USA

Email: aja0056@auburn.edu

Reviewer #2 (Remarks to the Author):

This manuscript describes an interesting and novel study of quasi Pd1Ni single-atom surface alloy catalyst to enable hydrogenation of nitriles to secondary amines without toluene. The catalysts were prepared by wet chemistry and Atomic Layer Deposition (ALD), and characterized by TEM, HAADF-STEM, XAS, XPS and DRIFTS. The authors performed theoretical calculations by way of DFT. After the revision, the experimental data of this manuscript is more sufficient and convincing. As this study open a new way to design metal

catalysts in selective hydrogenation of nitrile, I believe this manuscript can be accepted for publication in Nature Communication.

Response: We gratefully thank the reviewer's positive comments on our manuscript.

Reviewer #3 (Remarks to the Author):

The authors have addressed my previous comments and I recommend acceptance of the manuscript for publication in its current form.

Response: We gratefully thank the reviewer's constructive comments on our manuscript.